# Higher Infection Risk among Health Care Workers and Lower Risk among Smokers Persistent across SARS-CoV-2 Waves—Longitudinal Results from the Population-Based TiKoCo Seroprevalence Study

**DOI:** 10.3390/ijerph192416996

**Published:** 2022-12-17

**Authors:** Felix Günther, Sebastian Einhauser, David Peterhoff, Simon Wiegrebe, Hans Helmut Niller, Stephanie Beileke, Philipp Steininger, Ralph Burkhardt, Helmut Küchenhoff, Olaf Gefeller, Klaus Überla, Iris M. Heid, Ralf Wagner

**Affiliations:** 1Department of Mathematics, Stockholm University, Albanovägen 28, 11419 Stockholm, Sweden; 2Institute of Medical Microbiology and Hygiene, Molecular Microbiology (Virology), University of Regensburg, Franz-Josef-Strauß-Allee 11, 93053 Regensburg, Germany; 3Institute of Clinical Microbiology and Hygiene, University Hospital Regensburg, Franz-Josef-Strauß-Allee 11, 93053 Regensburg, Germany; 4Department of Genetic Epidemiology, University of Regensburg, Franz-Josef-Strauß-Allee 11, 93053 Regensburg, Germany; 5Statistical Consulting Unit StaBLab, Department of Statistics, LMU Munich, Geschwister-Scholl-Platz 1, 80539 Munich, Germany; 6Institute of Clinical and Molecular Virology, University Hospital Erlangen, Friedrich-Alexander-Universität Erlangen-Nürnberg (FAU), Schlossgarten 4, 91054 Erlangen, Germany; 7Institute of Clinical Chemistry and Laboratory Medicine, University Hospital Regensburg, Franz-Josef-Strauß-Allee 11, 93053 Regensburg, Germany; 8Department of Medical Informatics, Biometry and Epidemiology, Friedrich-Alexander-Universität Erlangen-Nürnberg (FAU), Waldstr. 6, 91054 Erlangen, Germany

**Keywords:** SARS-CoV-2, seroprevalence, population-based study, longitudinal, risk factors

## Abstract

SARS-CoV-2 seroprevalence was reported as substantially increased in medical personnel and decreased in smokers after the first wave in spring 2020, including in our population-based Tirschenreuth Study (TiKoCo). However, it is unclear whether these associations were limited to the early pandemic and whether the decrease in smokers was due to reduced infection or antibody response. We evaluated the association of occupation and smoking with period-specific seropositivity: for the first wave until July 2020 (baseline, BL), the low infection period in summer (follow-up 1, FU1, November 2020), and the second/third wave (FU2, April 2021). We measured binding antibodies directed to SARS-CoV-2 nucleoprotein (N), viral spike protein (S), and neutralizing antibodies at BL, FU1, and FU2. Previous infection, vaccination, smoking, and occupation were assessed by questionnaires. The 4181 participants (3513/3374 at FU1/FU2) included 6.5% medical personnel and 20.4% current smokers. At all three timepoints, new seropositivity was higher in medical personnel with ORs = 1.99 (95%-CI = 1.36–2.93), 1.41 (0.29–6.80), and 3.17 (1.92–5.24) at BL, FU1, and FU2, respectively, and nearly halved among current smokers with ORs = 0.47 (95%-CI = 0.33–0.66), 0.40 (0.09–1.81), and 0.56 (0.33–0.94). Current smokers compared to never-smokers had similar antibody levels after infection or vaccination and reduced odds of a positive SARS-CoV-2 result among tested. Our data suggest that decreased seroprevalence among smokers results from fewer infections rather than reduced antibody response. The persistently higher infection risk of medical staff across infection waves, despite improved means of protection over time, underscores the burden for health care personnel.

## 1. Introduction

SARS-CoV-2 antibodies measured in serum of population-based studies’ participants provide insights into the proportion of individuals who have experienced infection. SARS-CoV-2 infection-mediated seroprevalence has been reported to be increased for medical personnel and decreased for smokers in numerous studies [1,2,3,4,5,6,7,8,9,10], including two studies from Germany [11,12] as well as our baseline (BL) assessment of the Tirschenreuth Study (TiKoCo) [13]. With the exception of [12], these results were based on cross-sectional data collected shortly after the first pandemic wave (summer 2020) or in fall 2020.

Increased seropositivity among medical personnel during the first infection wave in Europe is readily explained by higher infection risk due to occupational exposure. The early pandemic increased infection risk for medical personnel could have been due to limited knowledge regarding transmission routes or limited availability of protective gear. Some early studies suggested that health care workers supplied with appropriate protective gear were not at increased risk for SARS-CoV-2 infection [14]. However, it is still an open question whether the medical staff was still at increased infection risk during the second/third wave after widespread introduction of full protective gear at medical workplaces.

The reasons for the association between smoking and decreased seropositivity are more elusive. In principle, such results could point towards a reduced risk of infection among smokers [15,16]. However, any underlying mechanisms for this remain unclear. An alternative explanation is a decreased antibody response after infection among smokers compared to non-smokers, in line with a suppressed immune system. Interactions between smoking and the immune system are widely acknowledged (e.g., [17,18,19]) and also hypothesized for immune responses to SARS-CoV-2 [7,20]. An evaluation of differences in antibody response by smoking status is lacking so far. Also lacking are longitudinal studies evaluating whether the smoking association with seropositivity was persistent over time.

Here, we set out to investigate whether the association between working in a medical occupation or smoking with seropositivity was persistent across infection waves. We also aimed to evaluate a possible link between smoking and the development of SARS-CoV-2 specific antibodies after infection or vaccination. For this, we conducted longitudinal analyses in our cohort study from the Tirschenreuth population, the hardest-hit county in Germany during the first SARS-CoV-2 wave. We analyzed the association of smoking and medical occupation status with seropositivity, registered infections, and antibody response after infection or vaccination for three observation periods: pre-pandemic to July 2020 (first wave), July 2020 to November 2020 (few infections), and November 2020 to April 2021 (second/third wave).

## 2. Materials and Methods

### 2.1. Study Design, Participants, and Setting

The study was designed as a cohort study of the population aged at least 14 years living in the Tirschenreuth county, Germany, as described previously [13]. Briefly, 6608 individuals aged at least 14 years, randomly selected via population registries, were invited to study centers (or to request house visits, if necessary). Of these, 4203 individuals participated and provided informed written consent (net baseline response 64.3%). The study was conducted according to the Declaration of Helsinki. Participants provided blood and a self-completion questionnaire at baseline (BL, between 28 June and 13 July 2020) and two follow-up examinations, FU1 (16 November–27 November 2020) and FU2 (19 April–30 April 2021) [21]. 

Local health authorities registered 1109 SARS-CoV-2 infections in the county until 4 July 2020, 513 additional infections until 18 November 2020, and 3100 additional infections until 21 April 2021 [21]. For this, we set the cut-off date for registered infection as the fourth day prior to the median day of the study period at BL, FU1, and FU2 (assuming time from first symptoms to seroconversion as 12 days [22] and from first symptom to registration as 8 days [23]). During the 1st wave, all work with personal customer contact was prohibited except for medical occupations and food service; schools and pre-schools were closed. Similar restrictions were implemented during the 2nd/3rd wave (allowing work also for hairdressers and schooling of graduation classes).

### 2.2. Smoking and Occupational Status Assessed by Questionnaire

Using the pre-sent, written questionnaire in German language, participants provided information on smoking and occupational status. Specifically, participants were asked at BL, FU1, and FU2 whether they were currently smoking cigarettes, have previously smoked, or have never smoked. Current smokers were asked as to how many cigarettes they were smoking, and ex-smokers about their age when stopping. If participants reported to be current or ex-smokers previously and later on to be never-smokers, they were classified as ex-smokers for the later time point. The smoking status in the association analysis reflected the status at blood draw. 

Participants were asked at BL whether they were employed in February 2020 and, if yes, in which occupation (in case of multiple employments, the occupation with >50% of their working time). The options were: (i) medical occupation in medical care, care home, or midwife, (ii) teacher at school or pre-school, (iv) service or cook in food service, (v) cashier or shop assistant at groceries, (vi) hairdresser, (vii) other service with customer contact, or (viii) any other. In our analysis on the association of occupation and N-seropositivity as proxy for previous infection, we were particularly interested in three occupation groups, namely medical staff, grocery workers, and teachers, as for those groups, an increased exposure to the SARS-CoV-2 virus in the work setting is well imaginable. Grocery workers were one of the few occupational groups who were working during the first SARS-CoV-2 wave despite lockdown measures and strong restrictions on work with personal customer contact. Teachers can be suspected to be at high exposure during the 2nd/3rd wave and medical staff potentially over the whole period of the pandemic. From this motivation and with the intention to avoid small subgroups, we classified the occupational status of the study participants into four distinct groups: the three mentioned as well as a fourth group of “others” including all other types of self-reported occupations as well as unemployed individuals. Note that all analyses regarding occupation are restricted to individuals of working age (20–69 years).

### 2.3. Assessment of Other Covariates by Questionnaire at BL

Via the questionnaire at BL, as reported previously [13,21], participants were asked about the number of people living in their household and the highest educational degree including university/vocational school. Years of education were summed across educational steps (set at 6 years for pupils) and dichotomized at ≥13 vs. <13 years; high education thus corresponded to at least a graduation at the highest school level in Germany (“Gymnasium”). From reported alcohol consumption frequency (never, ≤1 per month, 2–4 times per month, 2–4 times per week, 5–6 times per week, ≥1 per day) and number of drinks consumed on a typical drinking day (beer 0.33 l, wine 0.125 l, liquor 4 cl), we derived the average number of daily drinks and “high/low alcohol consumption” (≥2/<2 drinks per day). We defined “high/low physical activity” (≥2/<2 h per day). Body mass index was computed using self-reported height and weight.

### 2.4. Assessment of Known Infections and Vaccination Status among Participants

By questionnaire, participants were asked whether they had undergone a SARS-CoV-2 test by trained staff (at medical practitioner, test site, or hospital) and whether this test was positive: ever (BL), since 17 July 2020 (FU1), or since 27 November 2020 (FU2). Self-reported positive tests were validated by health authorities. At FU2, participants were also asked whether they were vaccinated against SARS-CoV-2 and, if yes, how often, which vaccine, and when. Individuals were classified as fully vaccinated, if they had two vaccinations by Comirnaty (BionNTech, Mainz, Germany), SpikeVax (Moderna, Cambridge, MA, USA), or Vaxzevria (AstraZeneca, Cambridge, UK) ≥ 14 days before their respective blood draw.

### 2.5. Blood Processing, Storage, and Serum Antibody Measurements

Sample processing and antibody measurements were reported previously [13,21,24]. Briefly, sampled whole blood was transported daily to the University of Regensburg and processed on the same day. We measured serum antibodies (i) directed at the SARS-CoV-2 nucleoprotein (N, Elecsys Anti-SARS-CoV-2, Roche Diagnostics GmbH, Penzberg, Germany), (ii) the receptor-binding domain (RBD) of the viral spike protein (S, Elecsys Anti-SARS-CoV-2, Roche Diagnostics GmbH, Penzberg, Germany), and (iii) neutralizing antibodies. Both Elecsys tests were operated on a COBAS pro e 801 module according to the manufacturer’s instructions. SARS-CoV-2 neutralization capacity was evaluated using Vesicular Stomatitis Virus (VSV–ΔG*FLuc) pseudotyped with SARS-CoV-2-Spike-ΔER. Herein, triplicates of a fixed inoculum of 25,000 ffu were neutralized for 1 h with a 2-fold serum dilution series starting at 1 in 20. Luciferase activity was determined 20 h post-infection of HEK293T-ACE2+ cells using BrightGlo (Promega Corp, Madison, WI, USA). IC50 values (50% maximal inhibitory concentration) were calculated by curve fitting in GraphPad Prism 8 software (GraphPad Software, San Diego, CA, USA). N-antibodies were measured for all individuals at all three time points, S-antibodies for all at FU2 and for N-positives at any time point, and neutralizing antibodies for N-positives at any time point and for vaccinated individuals at FU2.

Whereas S-specific binding and neutralizing antibodies offer insights into possible protection [25], neither can distinguish vaccination from infection. On the other hand, N-specific antibodies prove previous infection, within the limits of assay sensitivity and specificity and depending on the durability of antibody responses [21], since the nucleoprotein (N) is not part of the administered vaccine formulations and therefore, N-specific antibodies are not developed in reaction to the SARS-CoV-2 vaccination. 

Thus, for the main analysis, seropositivity from infection was judged by N-antibody levels (cutoff = 1.0 Arbitrary Units (AU)/mL): participants N-seronegative at BL and N-seropositive at FU1 were classified as newly seropositive at FU1, and participants N-seronegative at FU1 and N-seropositive at FU2 as newly seropositive at FU2. Antibody response after PCR-confirmed infection or after full vaccination was based on N-, S- (S-cutoff = 0.8 Binding Antibody Units (BAU)/mL), or neutralizing antibodies (cutoff Inhibitory Dose 50 (ID50) = 20, as performed in [24]). For the analysis of antibody response after a PCR-confirmed infection based on S- or neutralizing antibody measurements at FU2, we focused on individuals without any vaccination only.

### 2.6. Statistical Analyses

At each of the three observation periods, we analyzed the association of smoking or occupational status with (new) N-seropositivity. Participants were included in these analyses when they had an N-antibody measurement available at the respective time point (BL, FU1, FU2) and, for FU1 and FU2, when they had N-antibodies measured and negative at the previous time point. For the smoking association, we restricted this to participants with information on smoking status at the respective time point. For the occupation association, we focused on participants at working age (20–69 years old) and analyzed medical staff, preschool/school teachers, and grocery employees vs. any other (to avoid small further subgroups). We applied logistic regression (R package mgcv [26]) using two models: (1) adjusted for age (non-linear using thin-plate regression splines) and sex, and (2) additionally adjusted for socio-demographic and lifestyle factors as published previously [13] (only for BL and FU2; too few new seropositives at FU1). We evaluated the association of number of cigarettes smoked (Winsorized at 30) with (new) seropositivity. We compared associations for N-based seropositivity with S-based seropositivity (only in FU2 sera, where S-antibody measurements were available for all participants). 

For the smoking association, we conducted additional analyses: (i) we evaluated antibody response after infection (defined by previous health authority-validated positive PCR test) by smoking status: we analyzed seropositivity and median levels of N-, S-, and neutralizing antibodies at FU2 by smoking status. For the S- and neutralizing antibodies, we restricted this analysis to unvaccinated individuals to exclude distortions due to vaccine-induced antibodies. (ii) We evaluated fully vaccinated participants at FU2 analyzing S- and neutralizing antibody levels measured at FU2 by smoking status. (iii) We evaluated the association of antibody levels with age and time since vaccination using a generalized additive model (Gaussian model with log-link, non-linear associations with age and time since vaccination using thin-plate regression splines). We compared this with an extended model adding smoking status and its interaction with age and time since vaccination and tested the null hypothesis of “no association of smoking with expected antibody levels” (approximate F-test). 

Finally, we evaluated, for each of the three observation periods, the proportion of participants with a health authority-validated positive test among those tested by smoking status and medical occupation and the proportion of tested among all participants.

All statistical analyses were performed in the R environment for statistical computing, version 4.1.0 (R Foundation for Statistical Computing, Vienna, Austria). 

## 3. Results

### 3.1. Analyzed Participants

These analyses included 4181 participants of the longitudinal Tirschenreuth Study with valid measurements of N-antibodies at BL (n = 3513 or 3374 with measurements also at FU1 or FU2, respectively, n = 3177 with measurements at all three time points). At BL, participants were aged 14 to 102 years, 51.6% women, 20.4% current cigarette smokers, and 8.0% of the 3303 participants aged 20 to 69 years worked in medical occupation (Table 1). Response at FU1 and FU2 was >80%, as described previously [21], lower among the younger (e.g., 78% and 79% at FU1 and FU2 for 14–19 year-old vs. 87% and 84% for 70+) and among current smokers (77% and 71% vs. 87% and 84% among never-smokers), but not differential by occupation (Appendix A); the response was higher among N-seropositives than N-seronegatives at BL (FU1: 94% vs. 83%; FU2: 93% vs. 79%). 

SARS-CoV-2 vaccination roll-out in Germany started in late December 2020, yielding the first fully vaccinated individuals at the end of January. During the early phase of the vaccination program, individuals at increased risk of severe disease and individuals at increased risk of exposure were prioritized. As a consequence, we observed increased fractions of vaccinated individuals at FU2 in the higher age groups (fully vaccinated: 32% in 70+ vs. 5% in 14–69-year-olds, partly vaccinated: 54% in 70+ vs. 32% in 14–69-year-olds, Appendix A) as well as among medical personnel and teachers (fully vaccinated: 39%, 5%, 0%, 2% for medical, teacher, grocery, and other occupation with age 20–69 years; partly vaccinated: 28%, 54%, 28%, 33% for medical, teacher, grocery, and others, Appendix A). Of note, we did not find differences in vaccine uptake with respect to smoking status in our data (fully vaccinated: 9%, 10%, 9% for current, ex-, and never-smokers; partly vaccinated: 35%, 41%, 33% for current, ex-, and never-smokers, Appendix A).

### 3.2. New N-Seropositivity Was Decreased for Smokers and Increased for Medical Personnel in Each of the Three Observation Periods

For each of the three observation periods (BL after first wave, low incidence between FU1 and FU2, second/third wave between FU1 and FU2), we evaluated the association of smoking and occupation status with new N-seropositivity. New N-seropositivity was about half among current smokers compared to never-smokers for all three observation periods. 

Figure 1A: Age- and sex-adjusted odds ratios (OR) were 0.47 (95% confidence intervals, CI, = 0.33–0.66, *p* < 0.001), 0.40 (95%-CI = 0.09–1.81, *p* = 0.22), and 0.56 (95%-CI = 0.33–0.94, *p* = 0.02), respectively. Note that the estimate at FU1 shows large uncertainty, as the overall number of new seropositive cases was rather low, but the point estimate is in line with results from BL and FU2. The smoking association with N-seropositivity was not observed for ex-smokers compared to never-smokers (Figure 1A; *p* ≥ 0.05, Appendix A). 

The current smoking association was underscored by a dose–response association when analyzing the number of daily currently smoked cigarettes at BL and FU2 (BL: OR = 0.75 per 5 cigarettes a day, 95%-CI = 0.65–0.84, *p* < 0.001; FU2: OR = 0.78, 95%-CI = 0.63–0.93, *p* = 0.012, based on a logistic regression model with linear association of the log odds of new N-seropositivity with the number of cigarettes smoked; not analyzed at FU1, too few new seropositives). Appendix A shows a decrease in the model-based probability for new N-seropositivity in association with an increasing number of cigarettes smoked based on a generalized additive model with logistic link allowing for a potentially non-linear association with the number of cigarettes smoked.

For medical personnel, new N-seropositivity was doubled for the first wave period, also increased during the low-infection period in summer 2020, and threefold increased for the second/third wave period compared to other occupations (excluding grocery employees and preschool/school teachers (Figure 1B): age- and sex-adjusted ORs were 1.99 (95%-CI = 1.36–2.93, *p* < 0.001), 1.41 (95%-CI = 0.29–6.80, *p* = 0.67), and 3.17 (95%-CI = 1.92–5.24, *p* < 0.001), respectively (Appendix A). There was no significant difference in new N-seropositivity for grocery employees or preschool/school teachers compared to other occupations (*p* ≥ 0.05, Figure 1B, Appendix A).

When comparing the smoking and occupation associations with N-seropositivity as well as S-seropositivity at FU2, we found consistent age-/sex-adjusted ORs for both types of seropositivity (Appendix A).

### 3.3. Associations Were Robust upon Adjustment by Other Factors and across Subgroups

The observed associations of smoking and medical occupation with (new) seropositivity at BL and at FU2 (compared to FU1) remained when adjusted for each other and for age, sex, education, household size, physical activity, alcohol consumption, and BMI (Appendix A; FU1 not analyzed, too few N-seropositives). This was observed for all participants and when restricted to working age (20–69 years old).

The smoking association with (new) seropositivity at BL and at FU2 (compared to FU1) was consistent across subgroups (age < 50/50+, men/women, high/low education, chronic diseases yes/no, medical occupation yes/no; Appendix A). This indicated no confounding by or interaction with these factors, which are potentially linked to less smoking or less outside contact (e.g., higher education or chronic diseases). Specifically, we found a consistent smoking association with reduced N-seropositivity also among medical staff (11.4% vs. 18.5% new N-seropositives in current vs. never-smokers at BL, OR: 0.56 95%-CI: 0.25–1.29; 9.3% vs. 13.7% at FU2, OR: 0.65 95%-CI: 0.19–2.20). There was no evidence for an interaction between medical occupation and smoking status with regard to infection risk.

### 3.4. Similar Antibody Response to Infection or to Vaccination for Smokers and Non-Smokers

Since decreased seropositivity for smokers may have derived from a lack of antibody response among smokers, we explored this further in our data. 

First, we evaluated antibody response after infection by smoking status. For this, we focused on participants with health authority-validated positive PCR tests reported at any time (until FU2) to define “infected” individuals independent of antibody-based serostatus and analyzed levels of N-, S-, and neutralizing antibodies at FU2 (for S- and neutralizing antibodies: restricted to unvaccinated, i.e., participants without any reported vaccine dose). We found similar median antibody levels by smoking status (Figure 2). The proportion of seronegatives among these participants with documented positive PCR tests was not increased among current smokers compared to ex- or never-smokers (N: 9%, 10%, 0% for never, ex-, and current smokers, respectively; S: 5%, 4%, 0%; neutralizing: 8%, 9%, 9%). 

Second, we sought further support of the hypothesis that smokers’ antibody responses resembled those of non-smokers by evaluating response after vaccination as a general model for antibody response. We restricted this to participants fully vaccinated without previous infection (i.e., two vaccinations ≥ 14 days before FU2; N-seronegative at FU2). We found similar levels of S- and neutralizing antibodies at FU2 by smoking status (Figure 3A,C), also when accounting for age and time since full vaccination. Model-based estimates of S- and neutralization antibody levels (expected levels) document the decrease by time since full vaccination and by age [27], but no impact by smoking status (Figure 3B,D, F-test *p* = 0.274 and *p* = 0.266 for S- and neutralizing antibodies, respectively). Since time since full vaccination was relatively short (up to 100 days, German vaccination roll-out since 27 December 2020) and the dose was standardized, this is an ideal model for antibody response. 

Overall, we found no evidence that current smokers developed fewer quantitative antibodies after infection or vaccination or that smokers were more likely seronegative than ex-/never-smokers based on three different serological tests.

### 3.5. Fewer Infected among Current Smokers Than among Non-Smokers in Each of the Three Observation Periods

As a third piece of evidence, we analyzed the proportion of participants who had tested positive for SARS-CoV-2 infection among those who reported a test for the respective observation period by smoking or medical occupation status. Interpreting these proportions is challenged by the potentially varying testing intensity by smoking or medical occupation status and over time. We thus began by evaluating the fraction of tested participants for the three observation periods (i.e., until BL, between FU1 and FU2, and between FU1 and FU2): among all participants, the fractions tested were 12.1%, 27.2%, and 45.0%, respectively; this documented the increased testing intensity over time. These testing fractions were similar for current smokers, but medical staff were tested markedly more often (Figure 4A). 

Next, we evaluated the proportion of positive tests (i.e., health authority-validated) among tested participants by observation period: this proportion was substantially lower among current smokers than among never-smokers for each of the three observation periods (Figure 4B): age-/sex-adjusted OR = 0.36 (95%-CI = 0.15–0.85), 0.22 (95%-CI = 0.03–1.81), and 0.58 (95%-CI = 0.33–1.00), respectively (Appendix A). These ORs were in line with the above noted odds for new seropositivity, as was the lack of association for ex-smokers compared to never-smokers (*p* ≥ 0.05). Together with the similar testing intensity by smoking status, the lower proportion of positive tests among current smokers supported the idea of fewer infections among current smokers.

The observed increased testing intensity among medical occupations (see above) can distort the comparison of the proportion of tested positives among medical staff compared to other participants. In addition, the higher seropositivity for medical personnel is realistically attributable to more infections and requires no further support. Still, we compared the proportion of positive test reports among those tested between medical staff vs. “other”: we found little difference (Figure 4B; age- and sex-adjusted ORs = 1.09, 0.73, 1.52, *p* = 0.81, 0.69, 0.12, Appendix A). Thus, the proportion of positive tests among those tested does not reflect the increased odds of infection among medical occupation participants.

## 4. Discussion

Based on longitudinal data from our population-based study well through the third SARS-CoV-2 wave in spring 2021, we demonstrated persistent, substantially increased seropositivity among medical personnel during the first and the second/third waves, while seropositivity was reduced by about half in current smokers. 

The finding of increased seropositivity among medical staff can be attributed to the increased exposure to SARS-CoV-2 at the workplace and thus increased infection probability. Our data thus underscore that the increased infection probability for medical staff was still observable and substantial also in the second/third wave when full protective gear was available and installed. While further evaluations are necessary also for the infection waves based on SARS-CoV-2 Delta and Omicron variants, our results indicate that increased infection risk of medical staff was not solely an issue of the early pandemic and requires continued attention. Our conclusion of increased infection risk among health care workers results from the significantly higher frequency of infection-related antibodies in medical staff compared to other occupational groups. Occupational groups were assessed by self-report via questionnaire. The group of medical personnel was defined broadly as individuals who worked in medical care, in nursing homes, or as midwives. There can be considerable within-group heterogeneity in the extent of exposure and risk of infection depending on the respective medical field (e.g., intensive care, general practitioner), occupational role (physician, nurse, midwife), extent of patient contact, number of working hours, etc. Due to the population-based sample of our participants, the proportions of the specific medical subspecialties among participants roughly reflect the proportions in the medical staff population, and our effect estimates can be considered an average across these subspecialties. A detailed work history and meticulous quantification of patient contact would be warranted in future studies in order to quantify the occupational infection risk more specifically and to identify those at highest risk.

The finding of decreased seropositivity among smokers requires more detailed consideration of potential bias and possible reasons. The observed association of smoking with decreased seropositivity was strong, stable across timepoints and subgroups, persistent when adjusted for potential confounding factors, and underscored by a dose–response association. The observed association with odds ratios around 0.5 were similarly observed in previously published seroprevalence studies of different designs [1,2,3,4,5,6,7,8,9,10,11,12]. Altogether, this supports a genuine relationship between smoking and lower seropositivity. 

A possible bias can derive from differential participation with respect to smoking and the outcome of interest. We did observe reduced follow-up participation among current smokers compared to never-smokers, but both at a relatively high level with 77% vs. 87% participation at FU1 and 71% vs. 84% at FU2. Our outcome was new seropositivity among those never seropositive until the start of the respective observation period [21]. When comparing the proportion of new seropositives between smokers and non-smokers, this is unbiased under random dropout among smokers and non-smokers with respect to new seropositivity. If there were more dropouts among newly infected current smokers than among newly infected never/ex-smokers, e.g., due to a more severe COVID-19 disease among smokers preventing further participation in the study, this could lead to a bias towards the observed association. Since the serostatus of dropouts is unobserved, this cannot be ruled out in principle. However, since severe COVID-19 disease is relatively rare and age-dependent, we would not expect such a bias to yield a false association in the observed magnitude and stability across age groups. 

Lower seropositivity among smokers can derive from lower infection probability or from lower antibody response. In order to fully explain the strong association, antibody response for smokers would need to be substantially decreased, basically nulled for ~50% of smokers. In our data, smokers and never-smokers showed similar levels of antibody response after infection in a variety of antibody tests. For this analysis, we defined “infection” by having had a previous positive PCR test validated by health authorities, yielding 237 participants with validated infection independent of serostatus. This supports the hypothesis that the decreased seropositivity among smokers observed here and by others was not explained by differential antibody response after infection, but rather due to fewer infections. 

We also evaluated antibody levels after vaccination by smoking status in a further 287 participants under the notion that vaccination-induced antibody generation is a role model for infection-induced generation where the “exposure” dose is standardized and time since “exposure” is known. A recent meta-analysis [20] concluded that active smoking may negatively impact humoral response to COVID-19 vaccines, although the study-specific evidence was conflicting and pathophysiologic mechanisms remain elusive. Our population-based study does not provide evidence of mitigated vaccine-induced antibody response in current smokers vs. never-smokers. More and larger longitudinal study data are required to disentangle this puzzle. Further data are also necessary to understand the decline of vaccine- or infection-induced antibodies and a potentially differential decline by smoking status. However, in order to yield an association of smoking and reduced seropositivity as strong as that observed here and by others, with odds ratios around 0.5, such a smoking-related faster decline would need to be strong and rapid, e.g., to the extent of zero antibodies for at least 50% of smokers within 5 months, since our time interval between infection and blood draw was no more than 5 months. 

In contrast to the lack of any support for differential antibody response by smoking status, our data indicated lower infection probability for smokers when evaluating individuals who reported previous SARS-CoV-2 testing: comparing current with never-smokers, the odds of a positive PCR test among individuals having been tested was similarly reduced as the odds of seropositivity, and this was consistent across the three observation periods. Together with the fact that the probability of having been tested did not vary much by smoking status, the magnitude of the observed smoking association with both seropositivity and PCR-detected infection was in line with a ~50% decreased infection risk for smokers. 

Together, our data suggest that lower seroprevalence among smokers is due to lower infection risk. The reason for this is still elusive. This might be behavioral or biological. Decreased risk of infection among smokers can derive from a smoking-specific behavioral pattern, e.g., leaving indoor gatherings to smoke outside. However, there is no evidence that the association is modulated by socio-demographic or health-related factors: we found similar associations of smoking and seropositivity across age groups, between men and women, with high or low education, individuals with chronic disease or without, and among individuals in medical occupation or not. Our results imply that any behavioral pattern of smokers leading to reduced infection risk was stable across such subgroups and across infection waves, rendering this explanation less plausible. 

If the reason for the smoking association is biological, it might be smoking-related differential gene expression [28]. Transcriptomics data from a bronchial epithelial cell air-liquid interface model showed a significant, selective reduction of membrane ACE-2 expression following smoking exposure, which directly correlated with nicotine delivery [29]. This model also highlighted differential regulation of genes known to be involved during the viral internalization process. Complementary, genome-wide association studies revealed genetic loci associated with infection susceptibility [30,31]. Respective pathways might overlap with biological pathways modulated by smoking. For example, identified loci near *MUC4* and *MUC16* were supported by genetic variants that increased mucin expression in cilial lung tissue that might be protective for infection [31]. Cigarette smoking is known to be associated with increased expression of certain mucins in human airways, such as mucin 1 and 4 [32], mucin 5A and C [33] or MUC16 [34]. Early analysis in *MUC4*^−^/^−^ mouse models suggest a protective role of mucin 4 during coronavirus pathogenesis [35,36]. It is perceivable that the decreased infection probability for smokers stems from a pathway pinpointed by these or other genetic association signals. 

If substantiated, smoking will certainly never be a preventive option against infection due to the severe well-known effects on cancer and cardiac risk. Particularly *MUC4* overexpression has been associated with various types of cancer [37,38], which underscores the need for a rigorous evaluation of eventually infection-protective pathways for severe adverse effects. Nevertheless, unraveling the reasons for this apparently protective association of smoking with SARS-CoV-2 infection will provide important, potentially biological, insights and might help identify protective paths.

The strength of our study is the population-based longitudinal approach covering individuals aged 14 to 102 and the infection occurrence over more than 1 year. Further strengths are detailed participant information on socio-demographic and lifestyle factors as well as several types of antibody measurements. When using antibody status as an indicator of previous viral infection, it is important to consider the potential existence of vaccine-induced antibodies. In the case of SARS-CoV-2, this is particularly important as the early vaccination campaign had to deal with initially limited vaccine availability by prioritizing individuals at increased risk of severe disease and individuals at increased risk of exposure, such as medical staff. In line with this, we observed a larger proportion of medical staff fully vaccinated until April 2021 than other individuals at working age. Using N-specific seropositivity as a marker for a previous infection and restricting the analysis of S- or neutralization antibodies to unvaccinated individuals enabled the focus on infection-related antibodies for our association analyses without vaccination-related distortions.

Limitations of our study are typical to cohort studies, including non-perfect response at BL and FUs. However, our response was rather high with 64% at BL and >80% at FUs. The time between BL and FU1 assessments captured a period of low incidence with overall few new infections during summer 2020, yielding low power for detecting statistically significant associations with new seropositivity at this time point of our study. It is nevertheless interesting and noteworthy that we observed consistent, albeit not statistically significant associations at FU1. Another limitation is that infection occurrence was predominantly assessed based on serostatus with limited knowledge of the infection date or viral dose, as for other seroprevalence studies. Our analyses of individuals who had undergone community-based PCR testing allowed for insights independent of serological tests. A strength of seroprevalence studies is that they capture symptomatic as well as asymptomatic infections and are independent of the community testing strategy and its changes over time. The presented results allow for a comparison of risk factors for the winter 2020/21 infection wave with the first infection wave in spring 2020. Our data do not stretch into the current Omicron-induced infections. However, given the multiple infections on top of a mostly vaccinated population at the current time point, these previous pandemic results enable an isolated view of mainly single, first-time infections and might serve as a role model for SARS-CoV-2 infections in principle. Further results from studies with longer follow-up into the current pandemic situation are warranted, but will be substantially complicated by the mix of infections.

## 5. Conclusions

Altogether, our results are in line with the notion that smoking-related pathways—behavioral or biological—overlap with infection-protective pathways that eventually might inspire behavioral measures and also the development of novel therapeutics. Our results also highlight a continued high infection risk for medical personnel well into winter 2021 despite full protection gear in place.

## Figures and Tables

**Figure 1 ijerph-19-16996-f001:**
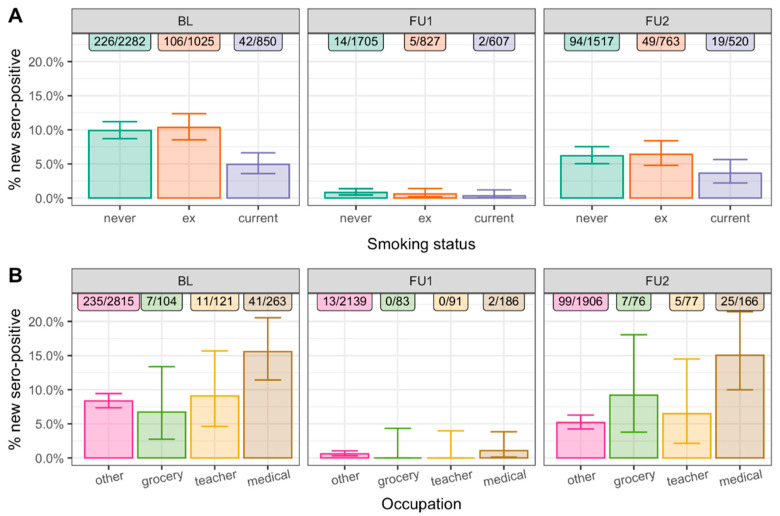
Proportion of N-seropositives by smoking status, occupation, and observation period. Shown are the proportions of new N-seropositives, number of N-seropositives, and number at risk for each of the three observation periods (BL, June 2020, BL to FU1, June–November 2020, FU1 to FU2, November 2020–April 2021): (**A**) for never, ex-, and current smokers (n = 4157 at BL) and (**B**) by occupation status (other, grocery employees, preschool/school teachers, medical staff) restricted to population at working age (20–69-year-olds, n = 3303 at BL). Whiskers represent 95% confidence intervals.

**Figure 2 ijerph-19-16996-f002:**
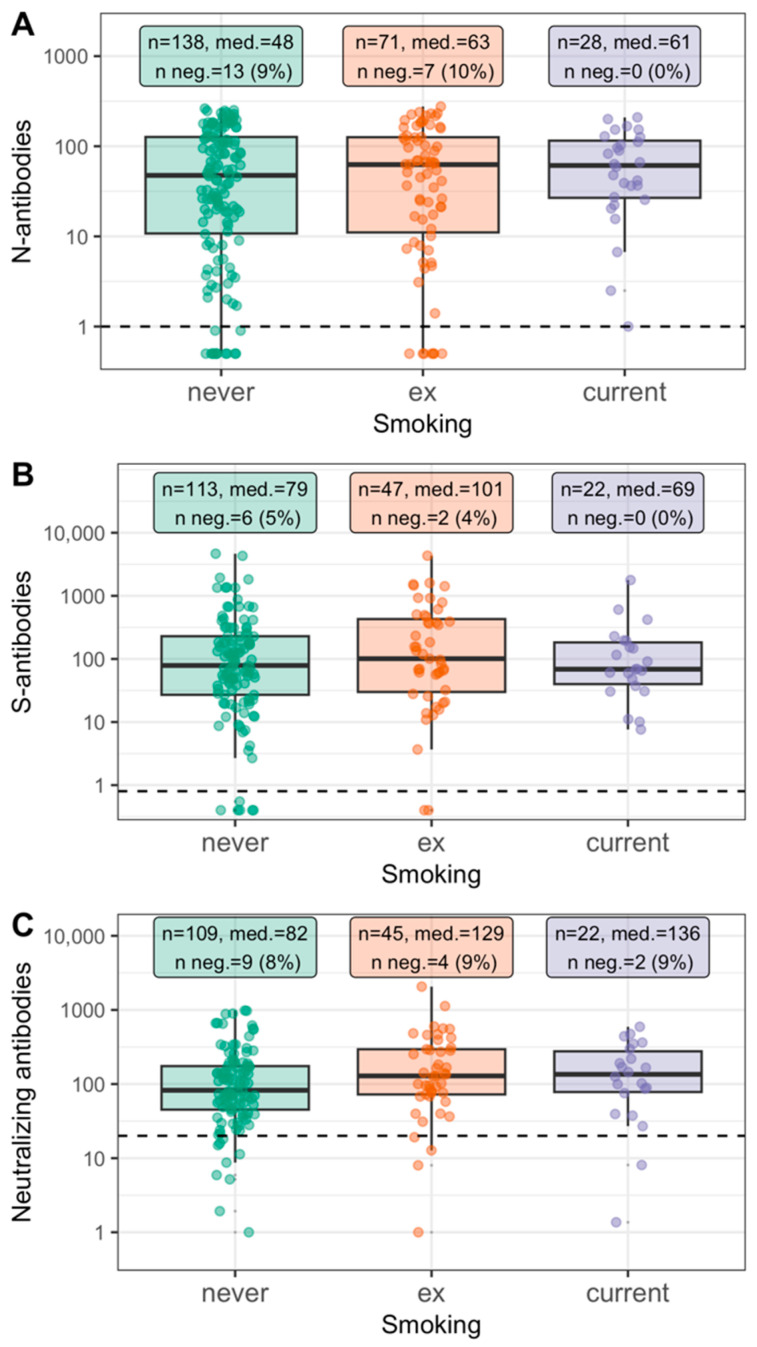
Antibody levels at FU2 among individuals with positive PCR test by smoking status. Shown are antibody levels at FU2 among individuals with positive PCR test in the study period (reported positive and health authority-validated, until April 2021, n = 237, including 28 current smokers): (**A**) N-specific antibodies, (**B**) S-specific antibodies (restricted to unvaccinated participants), (**C**) neutralizing antibodies (restricted to unvaccinated). Dotted line marks threshold for binary seropositivity. Indicated are sample size, median antibody level, number (%) of individuals seronegative for the respective antibodies. N-specific antibodies: Arbitrary Units (AU)/mL; S-specific antibodies: Binding Antibody Units (BAU)/mL; Neutralizing antibodies: Inhibitory Dose 50 (ID50).

**Figure 3 ijerph-19-16996-f003:**
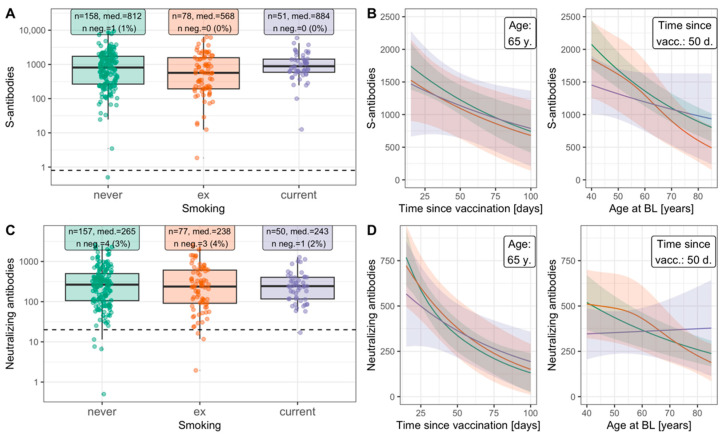
Quantitative antibody levels after vaccination by smoking status among uninfected. Shown are results regarding S-specific and neutralizing antibodies at FU2 among fully vaccinated individuals (i.e., second vaccination more than 14 days before blood draw) who were N-seronegative at FU2 (n = 287, including 51 current smokers). Panels (**A**,**C**): measured levels (dotted line = threshold for binary seropositivity), sample size, median antibody level, number (%) of individuals seronegative for the respective antibodies. Panels (**B**,**D**): expected antibody levels from a generalized additive model with covariates age (non-linear), time since vaccination (non-linear) and smoking status including interactions with age and time since vaccination. We did not find evidence for associations of expected antibodies with smoking status (Approximate F-test comparing this model to the corresponding model without smoking status and interactions: *p* = 0.274 and *p* = 0.266 for S-specific and neutralizing antibodies). S-specific antibodies: Binding Antibody Units (BAU)/mL; Neutralizing antibodies: Inhibitory Dose 50 (ID50).

**Figure 4 ijerph-19-16996-f004:**
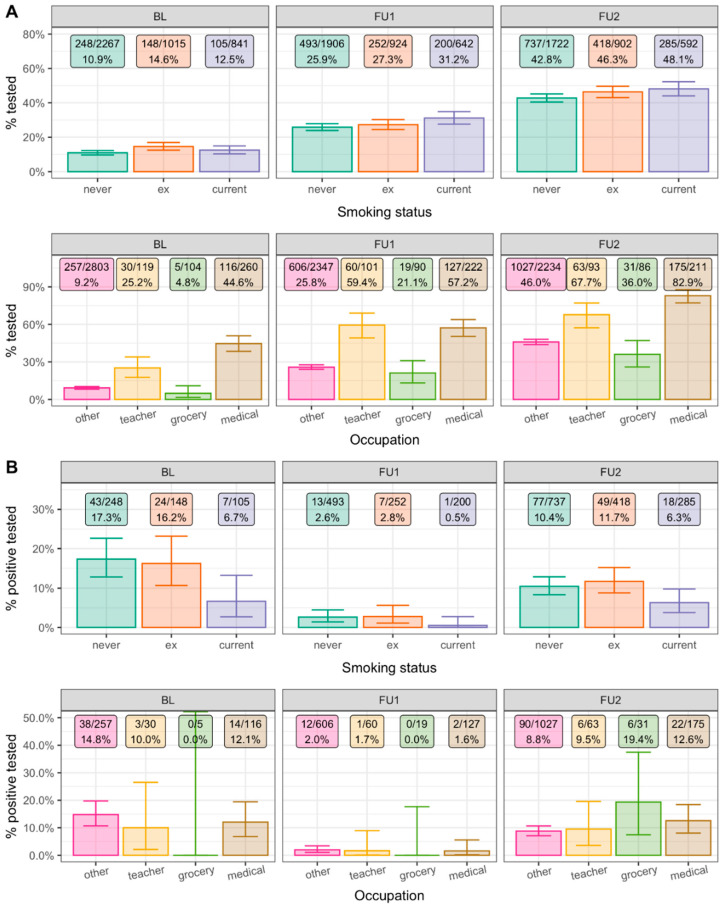
Fraction of tested and positive tested participants by smoking status and occupation. We evaluated the 4143, 3512, and 3337 participants with information on testing and test results for the three observation periods (BL, June 2020, BL to FU1, June–November 2020, FU1 to FU2, November 2020–April 2021), respectively. Shown are (**A**) the % tested and (**B**) the % tested positive (health authority-validated) by smoking status and occupation (other, grocery employees, preschool/school teachers, medical staff; restricted to working age, i.e., 20–69-year-olds). Whiskers represent 95% confidence intervals.

**Table 1 ijerph-19-16996-t001:** Participant characteristics at baseline and follow-up. For the participants at baseline (BL) and the two follow-ups (FU1, FU2), descriptive statistics are shown by characteristics (median and interquartile range, IQR, or proportions and absolute numbers) [and number of individuals with valid information]. All characteristics are self-reported, except N-seropositivity (i.e., proportion of positives for N-specific antibodies, cut-off = 1.0 (AU)/mL).

	BL [N = 4181]	FU1 [N = 3513]	FU2 [N = 3374]
**Age, sex (BL)**	**[N = 4181]**	**[N = 3513]**	**[N = 3374]**
Median age (IQR) [yrs]	52.0 (35.0–64.0)	53.0 (37.0–64.0)	53.0 (37.0–64.0)
Min, max age [yrs]	14.0, 102.0	14.0, 102.0	14.0, 102.0
Age 14–19 [yrs]: % (n)	5.4 (225)	5.0 (176)	5.2 (177)
Age 20–49 [yrs]: % (n)	40.8 (1707)	38.3 (1345)	38.1 (1284)
Age 50–69 [yrs]: % (n)	38.8 (1624)	41.2 (1449)	41.2 (1389)
Age 70+ [yrs]: % (n)	14.9 (625)	15.5 (543)	15.5 (524)
Women: % (n)	51.6 (2158)	53.0 (1861)	53.7 (1813)
**Chronic diseases**	**[N = 4081]**	**[N = 3435]**	**[N = 3300]**
Autoimmune: % (n)	7.1 (289)	7.3 (250)	7.4 (243)
Cancer: % (n)	4.9 (202)	5.2 (178)	5.0 (164)
Type 2 diabetes: % (n)	7.6 (312)	7.5 (259)	7.4 (245)
Cardiovascular: % (n)	9.9 (402)	9.6 (331)	9.5 (314)
None of these: % (n)	75.8 (3093)	75.6 (2596)	76.0 (2507)
**Education**	**[N = 4085]**	**[N = 3433]**	**[N = 3301]**
Median (IQR) [yrs]	11.0 (10.0–14.0)	11.0 (10.0–13.0)	11.0 (10.0–14.0)
≥13 yrs: % (n)	30.0 (1225)	29.5 (1013)	29.8 (985)
**Occupation** (20–69 years)	**[N = 3303]**	**[N = 2773]**	**[N = 2652]**
Curr. working (BL): % (n)	74.0 (2444)	73.8 (2046)	74.1 (1965)
Medical: % (n)	8.0 (263)	8.1 (224)	8.1 (216)
Education: % (n)	3.7 (121)	3.6 (101)	3.6 (95)
Grocery: % (n)	3.1 (104)	3.2 (90)	3.3 (87)
**Smoking**	**[N = 4157]**	**[N = 3493]**	**[N = 3356]**
Never smoking: % (n)	54.9 (2282)	56.7 (1981)	57.3 (1923)
Ex-smoker: % (n)	24.7 (1025)	24.6 (860)	24.6 (827)
Current smoker: % (n)	20.4 (850)	18.7 (652)	18.1 (606)
**Other lifestyle factors**			
Alc. drinks, daily: median (IQR)	0.2 (0.0–0.6) [N = 4049]	0.2 (0.0–0.6) [N = 3412]	0.2 (0.0–0.6) [N = 3280]
>2 alc. drinks, daily: % (n)	6.5 (262) [N = 4049]	6.3 (216) [N = 3412]	6.2 (204) [N = 3280]
BMI: Median (IQR)	26.6 (23.7–30.4) [N = 4134]	26.6 (23.7–30.3) [N = 3474]	26.6 (23.7–30.4) [N = 3339]
**N-seropositive (BL)**			
% (n)	8.9 (374) [N = 4181]	10.0 (351) [N = 3513]	10.3 (349) [N = 3374]
% (n), age 14–19 yrs	10.7 (24) [N = 225]	13.6 (24) [N = 176]	13.0 (23) [N = 177]
% (n), age 20–49 yrs	8.6 (146) [N = 1707]	10.0 (135) [N = 1345]	10.4 (133) [N = 1284]
% (n), age 50–69 yrs	9.3 (151) [N = 1624]	9.8 (142) [N = 1449]	10.3 (143) [N = 1389]
% (n), age 70+ yrs	8.5 (53) [N = 625]	9.2 (50) [N = 543]	9.5 (50) [N = 524]
% (n), never-smoker	9.9 (226) [N = 2282]	10.8 (213) [N = 1981]	11.0 (212) [N = 1923]
% (n), ex-smoker	10.3 (106) [N = 1025]	11.4 (98) [N = 860]	11.7 (97) [N = 827]
% (n) current smoker	4.9 (42) [N = 850]	6.1 (40) [N = 652]	6.6 (40) [N = 606]

## Data Availability

The data presented in this study are available on request from the corresponding author.

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
