# Peer review of "Higher Infection Risk among Health Care Workers and Lower Risk among Smokers Persistent across SARS-CoV-2 Waves—Longitudinal Results from the Population-Based TiKoCo Seroprevalence Study"

_ijerph, 2022, doi:10.3390/ijerph192416996_

Round 1

Reviewer 1 Report

In this manuscript, Felix Gunther et al analyzed the association of occupation and smoking with  SARS-CoV-2 seropositivity and found infection risk is higher in health care workers and lower in smokers, which is correspond with previous studies after the 1st pandemic wave of SARS-CoV-2. The increased seropositivity among medical workers during 1st infection wave could be explained by occupational exposure, which is caused by limited knowledge of transmission route or limited availability of protective gear. It is unknown whether these health workers are still at higher infection rate after the 1st wave. Although it has been reported the smokers had decreased seropositivity, the reasons are elusive. In order to answer these questions, the authors analyzed seropositivity among health care workers and smokers during different infection waves. Moreover, they also tested SARS-CoV-2 binding antibodies after infection or vaccination. The finding from this study suggested higher infection risk is continued in health care workers despite full protection gear. Meanwhile, the lower seropositivity in smokers is due to lower antibody response, but a result of lower infection rate, which could be related with smoking-specific behavior pattern or biological mechanisms, which is interesting and could be considered for novel therapeutics.

This study is very important for understanding and controlling higher infection rate in health care workers during SARS-CoV-2 pandemic and investigate mechanism or developing novel therapeutic against SARS-CoV-2 infection based on the lower infection risk in smokers.  

Specific comments:

This study analyzed seropositivity and antibody response among health care workers and smokers during different wave of SARS-CoV-2 pandemic and found higher infection risk in health care workers, but lower infection risk in smokers. The data was properly presented, results and conclusions are clearly discussed.

Reviewer 2 Report

Reliability score of the questionnaire is required

Reviewer 3 Report

Dear Editor in Chief

I read the manuscript entitled “Higher infection risk among health care workers and lower risk among smokers persistent across SARS-CoV-2 waves-longitudinal results from the population-based TiKoCo seroprevalence study” that was a very well conducted study on SARS-Cov-2 antibody response in different population subgroups. However, it is principally an epidemiological study in details (not my skill) and therefore, this manuscript is highly recommended to be peer-reviewed by an expert in the field of biostatistics.

Here are some issues as well:

1. Please explain the reasons for choosing N-, S-, and neutralizing antibody measurement (objective of detecting each antibody) at different time points/individuals for readers in the 2.5. section.

What is the difference between anti-S and neutralizing antibody measurement?

What was the measurement method to assess neutralizing antibodies? (Please also describe this in the text)

2. Please include the specific results obtained from smokers versus non-smokers both with medical occupation in the manuscript text (supplementary table 4) and discuss about it. Did differences (given as percentages and ORs in the supplementary table 4) were statistically significant?

3- Characteristics (year, volume, page, …) are incomplete for some references (e.g. references # 5, 9, and 25).

Best regards

Reviewer 4 Report

First of all, thank you for asking me to revise the manuscript entitled “Higher infection risk among health care workers and lower risk among smokers persistent across SARS-CoV-2 waves - longitudinal results from the population-based TiKoCo seroprevalence study” by Günther et al.

The aim of this study was to investigate whether the association between working in a medical occupation or smoking with seropositivity was persistent across infection waves in a population living in the Tirschenreuth county in Germany. The authors also aimed to evaluate a possible link between smoking and the development of SARS-CoV-2 specific antibodies after infection or vaccination.

The authors present a further analysis, mainly based on smoking habit, of a cohort of subjects from a previous investigation conducted by the same authors (Wagner R, Peterhoff D, Beileke S, et al. Estimates and Determinants of SARS-Cov-2 Seroprevalence and Infection Fatality 537 Ratio Using Latent Class Analysis: The Population-Based Tirschenreuth Study in the Hardest-Hit German County in Spring 538 2020. Viruses 2021; 13: 1118. https://doi.org/10.3390/v13061118).

The language is generally fluent and correct. It is noteworthy that a large number of participants was available from the start to the end of the study (4181, 3513, 3374). Furthermore, the verification of the immune response in smokers and non-smokers is well done, evaluating post-vaccine antibody response and post-infection antibody assessment ascertained by PCR. So the authors successfully refute a hypothesis reported in the introduction (Karachaliou et al., 2021; Ferrara et al., 2022), according to which the lower seropositivity in smokers might be explained because of a suppressed immune system.

Despite the strengths outlined, the manuscript presents a series of critical points which, in my opinion, compromise the acceptance of the manuscript in this current form.

 MAJOR ISSUES

The main limitation of this manuscript is giving excessive importance to the increased seropositivity of workers in the health sector, drawing risky, albeit plausible, conclusions, as it will be discussed here.

The study is lacking in many information regarding the medical workers. It is necessary to characterize this category much better by indicating the workplace (whether they are general practitioners, whether they work in hospital, in a clinic, in a care home,…), their role at work , the contact with patients,  etc.

The vaccination status is an aspect which is neglected, but it is particularly important when evaluating the seropositivity and the risk of infection. To assess the risk of infection it would be necessary to know the vaccination status in order to discern the two conditions which lead to seropositivity. If the authors have taken this into account, they should underline it in the text

 TITLE: the title correctly reflects the observation that there is a ”lower risk among smokers”. However it is misleading when the title stresses a “higher infection risk among health care workers” as, as stated previously, the article neglects the vaccination status of such workers. Furthermore the categories of workers are generally composed of small numbers of individuals (see Figure 1), so it does not allow to affirm such hazardous statement. Thus in my opinion the title should be modified.

RESULTS:

- It is appreciable that the authors have reported the data collected during the FU1, but the scarcity of the number of new positive cases in this period makes it hard to have results valid and comparable for “each of the three observation periods” as the authors wrote in the titles of the paragraphs in the Results section and all over the text. The little numbers imply that the results cannot be attributed to all the observation periods, or at least it is not possible to affirm it for the FU1. So it is an error to report the results as valid at “each of the three observation periods” (e.g.: paragraph 3.2, 3.5).

- The working categories included in the "Other" category are not well specified (what kind of jobs are included?) and it is not clear why the authors considered important to distinguish the grocery employees (n=104 at BL, 83 at FU1, 76 at FU2), and the teachers (n=121, 91, 77) from the other categories investigated in the questionnaire and grouped in "Other". Furthermore, the new seropositives number among grocery employees and teachers is too scarce to draw relevant comparison between such categories and the “Other” and “Medical” categories. So caution is needed when drawing conclusions about infection risk by comparing such small numbers.

- Lines 223-226: “The current smoking association was underscored by a dose-response when analyzing the number of daily currently smoked cigarettes at BL and FU2 (BL: OR=0.95 per 5 cigarettes a day, 95%-CI=0.92-0.97, P<0.001; FU2: OR=0.95, 95%-CI=0.91-0.99, P=0.011; not analyzed at FU1, too few new seropositives).” If a dose-response association has been underscored, please describe in the text the other groups of current smokers stratified by the N of cigarettes smoked a day that have been considered and provide all the ORs regarding each group of current smokers. Only the OR for 5 cigarettes a day is reported.

- All over the text, it is not clear if the seropositivity declared in FU2 excludes vaccinated subjects or not.

In fact, the absolute numbers in the Supplementary Tables 3 and 5 are not the same as the numbers in the text: the numerators and denominators in Figure 1 are different from the numerators and denominators in Supplementary Table 3 and 5. It’s deducible that the authors may have excluded vaccinated subjects, but in the article this part is poorly explained and little detailed. Consequently, it is not plausible to present the results of this section as the main result of the whole study (“Higher infection risk among health care workers ...”).

DISCUSSION:

- Lines 341-345: it is not possible to demonstrate a “persistent, substantially increased seropositivity among medical personnel“ at all three observation points (BL, FU1, FU2) due to the too few new seropositives at FU1. The lack of information regarding the vaccination status, necessary to evaluate the meaning of seropositivity, is a major issue. That is, without distinguishing vaccinated and unvaccinated individuals relevant data about seropositivity cannot be drawn.

- Lines 346-348: That “increased seropositivity among medical staff can be attributed to the increased exposure to SARS-CoV-2 at the workplace and thus increased infection probability” may be true, but more cautious language is needed. It is known that the healthcare environment is subject to greater viral diffusion, but in order to be able to affirm that the increased seropositivity is correlated to an increased risk of infection, it is necessary to prove it also considering vaccination data.

REFERENCES

Please revise the “References” section as some references are incorrectly reported. In particular, some lack the year of publication (e.g. [5], [9], [11]).

OTHER REMARKS

It is interesting that the title of each paragraph in the “Results” section summarizes the main result of the paragraph, as if to anticipate the relevant data. Please, verify that such way of giving titles to paragraphs is in line with the editorial norms.

MINOR REVISION

Grammar and style:

“#” is used as “number” in the tables. It is probably too informal for a scientific paper and “N” should be preferred instead.

Please correct or consider the following as indicated:

-          Line 43/44: “our data suggests” should be corrected in “our data suggest”.

-          138: maybe it would be better to specify “vaccinated against SARS-CoV-2”.

-          163: maybe it would be better to repeat “when they”, thus making the sentence “and, for FU1 and FU2, when they had N-antibodies”.

-          347: SARS-CoV-2.

-          453: “our data does not” should be corrected in “our data do not”.

-          454: omicron-induced.

-          Supplementary Table 1: in the "Age, Sex" category, Women is used while in all the other tables Female is used.

-          Supplementary Table 4: line 3: Education; Subgroup Sex: female.

-          Supplementary Table 5: line 3: 95%-confidence.

Round 2

Reviewer 4 Report

Thank you for revising the manuscript according to our comments.

We now consider the manuscript clearer and more complete. For this reason we can accept it in this updated form.